# Mixed permutation symmetry quantum phase transitions of critical three-level atom models

**Alberto Mayorgas[1]⋆, J. Guerrero[2,3] and Manuel Calixto[1,2]**

**1** Department of Applied Mathematics, University of Granada,
Fuentenueva s/n, 18071 Granada, Spain
**2** Institute Carlos I of Theoretical and Computational Physics,
University of Granada, Fuentenueva s/n, 18071 Granada, Spain
**3** Department of Mathematics, University of Jaen,
Campus Las Lagunillas s/n, 23071 Jaen, Spain

⋆ albmayrey97@ugr.es

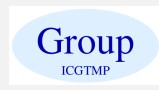
## Abstract

We define the concept of Mixed Symmetry Quantum Phase Transition (MSQPT), considering each permutation symmetry sector $\mu$ of an identical particles system, as singularities in the properties of the lowest-energy state into each $\mu$ when shifting a Hamiltonian control parameter $\lambda$. A three-level Lipkin-Meshkov-Glick (LMG) model is chosen to typify our construction. Firstly, we analyze the finite number $N$ of particles case, proving the presence of MSQPT precursors. Then, in the thermodynamic limit $N \to \infty$, we calculate the lowest-energy density inside each sector $\mu$, augmenting the control parameter space by $\mu$, and showing a phase diagram with four different quantum phases.



## 1 Introduction

When studying quantum systems of identical particles, e.g. bosons and fermions, permutation symmetry becomes crucial. A nontrivial example is that of $N$ identical particles distributed in a set of $L$ levels ($\mathcal{H}_L^{\otimes N}$ as Hilbert space) and a second quantized Hamiltonian describing pair correlations [1]. In particular, the condition of identical atoms allows to use permutation symmetry $S_N$ to decompose $\mathcal{H}_L^{\otimes N}$ into a "Clebsh-Gordan" direct sum of unitary irreducible representations (unirreps or sectors) of U($L$). We shall use Young tableaux as a useful graphical method to depict this decomposition.

It is common in the literature the restriction to the totally symmetric unirrep or sector when studying quantum phase transitions (QPTs) of critical quantum systems in the thermodynamic limit $N \to \infty$, like in Refs. [2–4], reducing Hilbert space $\mathcal{H}_L^{\otimes N}$ dimension from $L^N$ to, for

example, $\binom{N+L-1}{N} = N+1$ for $L = 2$. This means to make the particles indistinguishable, which is a broadly assumed procedure without any evident physical justification (usually for computational benefit). Therefore, we are devoted to study the role of these often disregarded mixed permutation symmetry sectors in this work. As a paradigmatic case, we will use the Lipkin-Meshkov-Glick (LMG) Hamiltonian for $L = 3$ levels (2), where $\lambda$ will be the control parameter used to detect critical phenomena (QPTs). The case $L = 2$ (see [5]) is not considered because all sectors can be reduced to the symmetric one, and the cases $L > 3$ provide an extra difficulty when minimizing the energy surface of the Hamiltonian. We address the reader to Ref. [6] for more information.

The organization of this article is the following; in Section 2 we focus on a simplified version of the Hamiltonian for $L = 3$ levels, and examine the numerical/exact lowest-energy state inside different permutation symmetry sectors for a finite number of particles $N$. In Section 3, we find mixed symmetry quantum phase transitions (MSQPTs) in the thermodynamic limit $N \to \infty$ using variational states. At the end, in Sec. 4, we give the conclusions.

## 2 The 3-level LMG model. U($L$) unirreps and QPT precursors

Models describing pairing correlations are usually described by a Hamiltonian in the second quantization form

$$H_L = \sum_{i=1}^{L}\sum_{\mu=1}^{N} \varepsilon_i c_{i\mu}^\dagger c_{i\mu} - \sum_{i,j,k,l=1}^{L}\sum_{\mu,\nu=1}^{N} \lambda_{ij}^{kl} c_{i\mu}^\dagger c_{j\mu} c_{k\nu}^\dagger c_{l\nu}, \tag{1}$$

where $c_{i\mu}$ ($c_{i\mu}^\dagger$) destroys (creates) a particle in the $\mu$ state of the level $i$. Precisely, there is a finite number $N$ of identical particles distributed over $L$ energy levels ($N$-fold degenerate). Pairs of particles are scattered between the $L$ levels when considering the two-body residual interactions of strength $\lambda$, so that the total number of particles remains constant.

In our case, we focus on $L = 3$ level systems and apply the following list of restrictions to the Hamiltonian (1): Firstly, we define U($L$) generators as $S_{ij} = \sum_{\mu=1}^{N} c_{i\mu}^\dagger c_{j\mu}$ according to the Jordan-Schwinger map [7,8]. Secondly, we disregard interactions for particles in the same level and consider equal interactions in different levels, i.e. $\lambda_{ij}^{kl} = \frac{\lambda}{N(N-1)}\delta_{ik}\delta_{jl}(1-\delta_{ij})$. Thirdly, we transform the Hamiltonian into an energy density (intensive quantity) by separating the interaction strength $\lambda$ by the total particle pairs $N(N-1)$. Fourthly, we place the levels symmetrically about the level $i = 2$, $\varepsilon_3 = -\varepsilon_1 = \epsilon/N$ and $\varepsilon_2 = 0$. Eventually, the Hamiltonian turns into the 3-level simplified version of the LMG Hamiltonian,

$$H_3 = \frac{\epsilon}{N}(S_{33} - S_{11}) - \frac{\lambda}{N(N-1)}\sum_{i\neq j=1}^{3} S_{ij}^2. \tag{2}$$

It can be regarded as an extension of the paradigmatic $L = 2$ levels LMG Hamiltonian used in the shell model [9,10].

An interesting property of the 3-level LMG Hamiltonian (2) is that the $\lambda$-interaction only scatters pairs of particles, and therefore, conserves the parity $\Pi_i = \exp(i\pi S_{ii})$ of the population $S_{ii}$ in each level $i = 1, 2, 3$. Consequently, the parity symmetry is described by the parity group $\mathbb{Z}_2 \times \mathbb{Z}_2 \times \mathbb{Z}_2$ with the constraint $\Pi_1 \Pi_2 \Pi_3 = (-1)^N$. This symmetry will be spontaneously broken in the thermodynamic limit $N \to \infty$ leading to a highly degenerated ground state [11]. In addition, if we choose basis vectors adapted to irreducible representations of the Lie group $U(3)$, the Hamiltonian matrix (2) will be block diagonal, and hence the procedure presented in the following paragraphs.

We want to focus on the decomposition of the $N$-fold tensor product Hilbert space $\mathcal{H}_L^{\otimes N}$ of $N$ $L$-level atoms into U($L$) unirreps. In particular, we shall use Young tableaux and Gelfand-Tsetlin (GT) patterns along this article since they are powerful diagrammatic methods (see [7, 12] for more details and definitions). The fundamental $L \times L$ representation of U($L$) is given by a Young box $\square$, and states of one particle by Weyl patterns/tableaux, $\boxed{1} = |1\rangle$, $\boxed{2} = |2\rangle$, $\boxed{3} = |3\rangle$, ... In the case $L = 3$, we apply the Gram-Schmidt orthonormalization procedure to the columns of a complex triangular matrix $T$ in order to obtain unitary matrices of U(3)

$$T = \begin{pmatrix} 1 & 0 & 0 \\ \alpha & 1 & 0 \\ \beta & \gamma & 1 \end{pmatrix} \xrightarrow{\text{G-S}} V = \begin{pmatrix} \frac{1}{\sqrt{\ell_1}} & \frac{-\bar{\alpha}-\gamma\bar{\beta}}{\sqrt{\ell_1\ell_2}} & \frac{-\bar{\beta}+\bar{\alpha}\bar{\gamma}}{\sqrt{\ell_2}} \\ \frac{\alpha}{\sqrt{\ell_1}} & \frac{1+\beta\bar{\beta}-\alpha\gamma\bar{\beta}}{\sqrt{\ell_1\ell_2}} & \frac{-\bar{\gamma}}{\sqrt{\ell_2}} \\ \frac{\beta}{\sqrt{\ell_1}} & \frac{\gamma-\beta\bar{\alpha}+\gamma\alpha\bar{\alpha}}{\sqrt{\ell_1\ell_2}} & \frac{1}{\sqrt{\ell_2}} \end{pmatrix}, \tag{3}$$

which is parameterized by the complex parameters $\alpha, \beta, \gamma \in \mathbb{C}$, where $\ell_1 = |T^\dagger T|_1 = 1 + \alpha\bar{\alpha} + \beta\bar{\beta}$ and $\ell_2 = |T^\dagger T|_2 = 1 + \gamma\bar{\gamma} + (\beta - \alpha\gamma)(\bar{\beta} - \bar{\alpha}\bar{\gamma})$. Actually, the addition of the three Cartan phases $u_j = e^{i\theta_j} \in U(1), j = 1, 2, 3$ completes the parameterization as $U = V \cdot \text{diag}(u_1, u_2, u_3) \in U(3)$. This parameterization is chosen for convenience and is derived from the Bruhat decomposition, which is a general version of the Gauss-Jordan elimination and is related to the Schubert cell decomposition of flag manifolds [13]. However, there are many others relevant parameterizations in the field of spin coherent states such as [14–17].

The $L^N$-dimensional Hilbert space $\mathcal{H}_L^{\otimes N}$ is represented by the $N$-fold tensor product representation $\square \otimes \overset{(N)}{\cdots} \otimes \square$. The Hilbert space is reducible into invariant subspaces, which are graphically represented by Young frames of $h_1 + \cdots + h_L = N$ boxes labeled by $h = [h_1, \ldots, h_L]$, where $h_i$ is the number of boxes in a row $i = 1, \ldots, L$, fulfilling $h_1 \geq \cdots \geq h_L$.

We shall remind that Weyl patterns symbolize the different vectors of a given representation (Young frame). They are in semistandard form when labels (numbers) inside the pattern increase from the right to the left, and strictly increase from the top to the bottom. An important result is that the number of semistandard form Weyl patterns is the dimension of the unirrep. Another useful definition is the weight of a Weyl pattern, which is the vector $w = (w_1, \ldots, w_L)$ whose components $w_k$ are the population of level $k$, with $w_1 + \cdots + w_L = N$. The lexicographical rule states that a state of weight $w$ has lower weight than another with weight $w'$ if the first non-zero coefficient of $w - w'$ is positive. Notably, the highest weight (HW) vector of a unirrep $h = [h_1, h_2, h_3]$ of $U(3)$ is $w = (h_1, h_2, h_3)$.

The semistandard form Weyl patterns are in one-to-one correspondence with Gelfand-Tsetlin (GT) patterns [7], another useful diagrammatic method to express the vectors spanning U($L$) unirreps. GT patterns are labeled by vectors $|\mathrm{m}\rangle$, and are useful for obtaining the eigenvalues and matrix elements $\langle \mathrm{m}|S_{ij}|\mathrm{m}'\rangle$ of the collective operators $S_{ij}$ in each unirrep $h$. This is called the Gelfand-Tsetlin method [18, 19].

From this point on, we shall study the symmetry classification of the LMG U(3) Hamiltonian (2) eigenstates, and some QPT precursors. The free LMG U(3) Hamiltonian is obtained by taking $\lambda = 0$ in (2), $H^{(0)} = \frac{\epsilon}{N}(S_{33} - S_{11})$, $\epsilon > 0$. According to the Lieb-Mattis theorem [20, 21], the lowest-energy eigenstate is the highest weight vector of the fully symmetric unirrep $h = [N, 0, 0]$, which corresponds to arrange all the particles in the level $i = 1$, $|\psi_0\rangle = |\mathrm{m}_{\text{hw}}\rangle = \boxed{1 \cdots 1}$ ($N$ boxes). The excited states have an energy $E_n = \frac{n-N}{N}\epsilon$, $n = 1, \ldots, 2N$, and are highly degenerated, except for $E_0$ and $E_{2N}$. For instance, the states $\boxed{1 \cdots 1 \, 2}$ and $\begin{array}{l}\boxed{1 \cdots 1}\\ \boxed{2}\end{array}$.

The two-body interactions governed by $\lambda$ lift the degeneracy of the eigenstates. For instance, the lowest energy in the unirrep $h = [3, 1, 0]$ is below the third lowest energy in $h = [4, 0, 0]$ for $\lambda < 1$, hence mixed symmetry sectors (such as $h = [3, 1, 0]$) should not

be disregarded in general when studying excited states and their energies.

At this point, it is convenient to define the concept of Mixed Symmetry Quantum Phase Transition (MSQPT) in a nutshell. We want to analyze critical behavior into each Hilbert subspace $\mathcal{H}_h$ corresponding to a unirrep $h$ of $U(3)$, as Hamiltonian evolution does not mix different sectors $h$. Consequently, we choose the lowest-energy vector $|\psi_0^h\rangle$ inside each $\mathcal{H}_h$, and seek abrupt changes in its structure when shifting $\lambda$ in the thermodynamic limit $N \to \infty$. But before doing that, we should consider QPT precursors for finite $N$ (exact eigenstates), which are calculated with exact/numerical Hamiltonian eigenstates and can anticipate the approximate situation of critical points. One of them is the fidelity [22, 23], measuring how similar (overlap) two states are in the vicinity ($\delta\lambda \ll 1$) of $\lambda$, $F_\psi(\lambda, \delta\lambda) = |\langle\psi(\lambda)|\psi(\lambda+\delta\lambda)\rangle|^2$. The fidelity reaches a minimum in the proximity of a critical point $\lambda^{(0)}$, when the state $|\psi(\lambda)\rangle$ suffers a drastic change of its structure. Another precursor, which is less sensitive to the step size $\delta\lambda$, is the susceptibility

$$\chi_\psi(\lambda, \delta\lambda) = 2\frac{1 - F_\psi(\lambda, \delta\lambda)}{(\delta\lambda)^2}, \tag{4}$$

which reaches a maximum in the vicinity of the critical point $\lambda^{(0)}$.

Figure 1a shows the susceptibility of the exact/numerical ground state (GS) of the LMG $U(3)$ model for different number of particles $N$. We have done the calculations numerically, giving a matrix form to the $S_{ij}$ operators using the GT basis $|\mathrm{m}\rangle$ in each unirrep. In particular, thanks to the Lieb-Mattis theorem [20,21], we know that the GS belongs to the fully symmetric irrep, reducing the computations to $h = [N, 0, 0]$ in this case. The susceptibility is sharper as $N$ increases, predicting a critical point around $\lambda \simeq 0.55\epsilon$ for the highest $N = 100$ curve, which is a precursor of the QPT eventually occurring exactly at $\lambda^{(0)} = 0.5\epsilon$ as we will see in Section 3.

On the other hand, Figure 1b displays the susceptibility of the exact lowest-energy vector inside different mixed symmetry sectors (unirreps $h$) for a fixed number of particles $N = 30$. Now, the would-be critical points (maximum of the susceptibility) move along the different sectors; they shift to the right from $h = [30, 0, 0]$ to $h = [20, 10, 0]$ (cyan dashed line), and to the left from $h = [20, 10, 0]$ to $h = [15, 15, 0]$ (magenta dashed line). Consequently, the figure envisages a quadruple point at the unirrep $h = [2N/3, N/3, 0]$. The maxima at the right in the figure are precursor of another QPT at $\lambda \simeq 1.5\epsilon$, but it is in a different scale and requires a higher $N$ to be properly characterized.

## 3 Thermodynamic limit and MSQPTs

We shall start this section talking about coherent states. They are excellent variational (semi-classical) states, as they reproduce the structure and mean energy density of lowest-energy states inside each symmetry sector $h$ at $N \to \infty$. For a detailed explanation, see the reference [24], and [5] for the U(2) case. In our case, we follow the Perelomov's construction [25, 26] of the coherent states in a given unirrep $h$ of U($L$). Namely, we rotate the HW vector state $|\mathrm{m}_{\mathrm{hw}}\rangle$ of a unirrep $h$ by a unitary matrix $U \in$ U(3) parameterized as in (3), $|h, U\rangle = K_h(U)|h; \alpha, \beta, \gamma\}$, where $|h; \alpha, \beta, \gamma\} = e^{\beta S_{31}} e^{\alpha S_{21}} e^{\gamma S_{32}}|\mathrm{m}_{\mathrm{hw}}\rangle$, and $K_h(U)$ is a normalization factor. For the totally symmetric unirrep $h = [N, 0, 0]$, the highest weight state is invariant under a U(2) subgroup, thus, any one of the exponential factors can be eliminated to properly define a U(3) CS. The coherent state expectation values $s_{ij} = \langle h, U|S_{ij}|h, U\rangle$ of the basic symmetry operators $S_{ij}$ can be easily calculated in the differential representation (see the Appendix A of [6] for a detailed calculation).

From now on, it is convenient to relabel U(3) unirreps $h = [h_1, h_2, h_3]$ by parameters $\mu, \nu$ (we only need two parameters because of the constraint $h_1 + h_2 + h_3 = N$.). More ex-

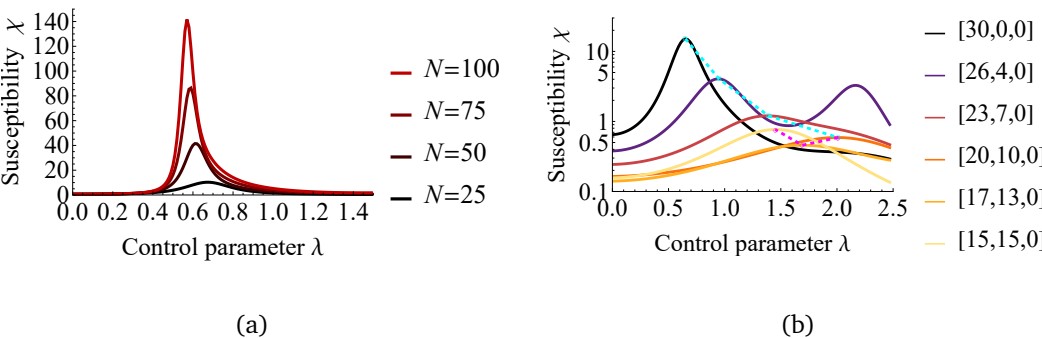

Figure 1: (a) Susceptibility $\chi_\psi$ of the ground state $\psi_0^{h=[N,0,0]}$ of the 3-level LMG Hamiltonian (2) for different values of the control parameter $\lambda$ and the total number of particles $N$. It predicts a QPT whose critical point is around $\lambda^{(0)} \simeq 0.55$. (b) Susceptibility $\chi_{\psi_0^h}$ (in logarithmic scale) of the lowest-energy vector $\psi_0^h$ into different sectors $h$ for a fixed number of $N = 30$ atoms. The dashed lines interpolate between the maxima of the susceptibilities, which are precursors of the would-be critical points dividing phase I from phase II (cyan), and phase I from phase IV (magenta) (see later on Figure 2 for the different phases). The turn away point, where both dashed lines meet, corresponds to the unirrep $h = [20, 10, 0]$, where four phases will coincide (see later on Sec. 3). We use $\epsilon$ units for $\lambda$ and a step size $\delta\lambda = 0.01$ in both figures.

plicitly, $h_3 = \nu N$, $h_2 = (1 - \mu)(1 - \nu)N$, $h_1 = \mu(1 - \nu)N$, for all $\nu \in [0, \frac{1}{3}]$, $\mu \in [\frac{1}{2}, \frac{1 - 2\nu}{1 - \nu}]$, becoming continuous parameters in the thermodynamic limit. Then, we are able to define the energy surface of a Hamiltonian density $H$ into the Hilbert space sector $(\mu, \nu)$ as $E_{\mu,\nu}^U(\epsilon, \lambda) = \lim_{N \to \infty} \langle h, U | H | h, U \rangle$. That is, the coherent state expectation value of the Hamiltonian density in the thermodynamic limit ($N \to \infty$). In the LMG U(3) case,

$$E_{\mu,\nu}^U(\epsilon, \lambda) = \lim_{N \to \infty} \left( \frac{\epsilon(s_{33} - s_{11})}{N} - \frac{\lambda \sum_{i \neq j=1}^{3} s_{ij}^2}{N(N-1)} \right), \tag{5}$$

which depends on the type of unirrep $(\mu, \nu)$, the complex coordinates of $U$ ($\alpha, \beta$ and $\gamma$), and the control parameters $\epsilon$ and $\lambda$. We fix $\epsilon$ and measure the energy surface and $\lambda$ in $\epsilon$ units, since $E_{\mu,\nu}^U(\epsilon, \lambda) = \epsilon E_{\mu,\nu}^U(1, \lambda/\epsilon)$. In addition, we benefit from $h = [h_1, h_2, h_3]$ and $h' = [h_1 - h_3, h_2 - h_3, 0]$ being equivalent SU(3) unirreps and obtain the expression $E_{\mu,\nu}^U(\epsilon, \lambda) = (1 - 3\nu)E_{\tilde{\mu},0}^U(\epsilon, (1 - 3\nu)\lambda)$, $\tilde{\mu} = \frac{\mu(1-\nu)-\nu}{1-3\nu}$, so we restrict to the study of the parent case $\nu = 0, \mu \in [\frac{1}{2}, 1]$. For $\mu = 1$, we have the totally symmetric representations, with a four-dimensional phase space $\alpha, \beta \in \mathbb{C}$ and an energy surface

$$E_{1,0}^{(\alpha,\beta)}(\epsilon, \lambda) = \epsilon \frac{\beta\bar{\beta} - 1}{\alpha\bar{\alpha} + \beta\bar{\beta} + 1} - \lambda \frac{\alpha^2(\bar{\beta}^2 + 1) + (\beta^2 + 1)\bar{\alpha}^2 + \bar{\beta}^2 + \beta^2}{(\alpha\bar{\alpha} + \beta\bar{\beta} + 1)^2}, \tag{6}$$

which is invariant under $\alpha \to -\alpha$, $\beta \to -\beta$, thus preserving the discrete parity symmetry inherited from the Hamiltonian). For $\mu = 1/2$, the representations are linked to rectangular Young tableaux ($h_1 = N/2 = h_2$), and the energy surface $E_{\frac{1}{2},0}^U(\epsilon, \lambda) = \frac{1}{2}E_{1,0}^{(\gamma,\beta')}(\epsilon, \frac{\lambda}{2})$, $\beta' = \beta - \alpha\gamma$, can be obtained from the totally symmetric case. The intermediate values $\mu \in (\frac{1}{2}, 1)$ give a six-dimensional phase space (flag manifold structure [13]) $\alpha, \beta, \gamma \in \mathbb{C}$, whose explicit energy surface expression is bulky.

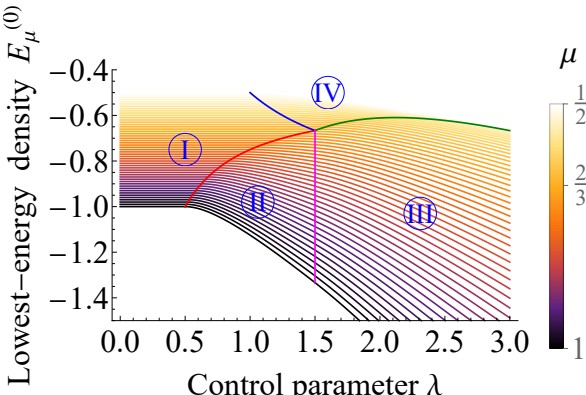

Figure 2: Lowest-energy density $E_\mu^{(0)}(\epsilon, \lambda)$ of the parental case (7) for different values of the control parameter $\lambda$ and the unirrep continuous parameter $\mu$, varying from $\mu = 1$ (black curve) to $\mu = 1/2$ (light yellow curve), with a step size of $\delta\mu = 0.01$. There are four different quantum phases in the phase diagram, which coincide at a quadruple point $(\lambda, \mu)_q = (3/2, 2/3)$. The phases are separated by curves of critical points in color red, magenta, green, and blue. Both axes are in $\epsilon$ units.

Henceforward we minimize in the phase space coordinates the energy density of the parent case (take $\nu = 0$, $\mu \in [\frac{1}{2}, 1]$ in (5)), i.e. we find the minimum energy

$$E_\mu^{(0)}(\epsilon, \lambda) = \min_{U \in U(3)} E_{\mu,0}^U(\epsilon, \lambda), \quad \forall \mu \in \left[\frac{1}{2}, 1\right]. \tag{7}$$

As we can see in Figure 2, the representation label $\mu$ behaves as an additional control parameter, differentiating four different quantum phases (I, II, III and IV) in the $\lambda$-$\mu$ plane (color lines). The transitions between phases for $\mu \neq 1$ can be understood as MSQPTs. We can also find the aforementioned quadruple point at $(\lambda, \mu)_q = (3\epsilon/2, 2/3)$, where the four phases coexist. The MSQPTs are second-order phase transitions as the second derivatives $\partial_{\mu\mu} E_\mu^{(0)}(\epsilon, \lambda)$, $\partial_{\lambda\lambda} E_\mu^{(0)}(\epsilon, \lambda)$, and $\partial_{\mu\lambda} E_\mu^{(0)}(\epsilon, \lambda)$ are discontinuous at critical points.

The minimization gives many critical points $\alpha_0, \beta_0, \gamma_0$ in the phase space with the same $E_\mu^{(0)}$, so the lowest-energy state for a general $\mu$ is highly degenerated. This behavior is easier to show in the fully symmetric case $\mu = 1$ (lowest lines of Figure 2), where there are three different phases and two second-order QPTs at $\lambda_{I\leftrightarrow II}^{(0)} = \epsilon/2$ and $\lambda_{II\leftrightarrow III}^{(0)} = 3\epsilon/2$. The critical values of $\alpha$ and $\beta$ which make the energy surface minimum are real numbers which have the properties $\alpha_0^\pm(\epsilon, \lambda) = 0$ $\forall 0 \leq \lambda \leq \frac{\epsilon}{2}$, and $\beta_0^\pm(\epsilon, \lambda) = 0$ $\forall 0 \leq \lambda \leq \frac{3\epsilon}{2}$ (check the reference [6] for an explicit expression of the minimum energy surface and the critical points). Therefore, there is a single minimum in phase I, $0 \leq \lambda/\epsilon \leq 1/2$, located at $\alpha = \beta = 0$; a double minimum in phase II, $1/2 \leq \lambda/\epsilon \leq 3/2$,, with $\beta = 0$; and a quadruple minimum in phase III, $\lambda/\epsilon \geq 3/2$. This degenerated minima effect is due to the spontaneous breakdown of the discrete parity symmetry of the Hamiltonian, as in the limit $N \to \infty$, the four coherent states $|\alpha_0^\pm, \beta_0^\pm\rangle$ reach the same minimum energy $E_1^{(0)}$ (minimization of the symmetric case $\mu = 1$, $\nu = 0$ in (5)). The parity restoration of the GS is discussed in the references [27, 28].

## 4 Conclusion

QPTs research in many-body systems usually presuppose the particle indistinguishability, restricting the scope to the fully symmetric representation ($\mu = 1$), which is often not a general

procedure. That is why we have defined MSQPTs as QPTs of the lowest-energy state in a particular symmetry sector $\mu$. As a test model, we have chosen an extension of the ubiquitous LMG model to $L = 3$ levels.

Firstly, we have done numerical calculations for a finite number of particles $N$ to obtain QPT precursors, such as the susceptibility, which anticipate the QPT in the thermodynamic limit $N \rightarrow \infty$. In general, the precursors give a better approximation to the critical points when increasing $N$.

Secondly, using coherent (semiclassical) states, we have considered the thermodynamic limit $N \rightarrow \infty$ and minimized the energy surface in different unirreps. The critical points $\lambda^{(0)}$, where the MSQPTs occur, turn out to depend on the representation index $\mu$. Therefore, we have extended the phase diagram in an extended control parameter space $(\lambda, \mu)$. In addition, there are evidences of a quadruple point where four different phases coincide at $\mu = 2/3$. We have also discussed that the lowest-energy state for general representation $\mu$ is degenerated, because of the spontaneous breakdown of the discrete parity symmetry of the Hamiltonian in the limit $N \rightarrow \infty$.

To conclude, we propose for further research the possible overlap between MSQPT and ESQPT [29], and the exploitation of permutation symmetry in the realm of quantum technologies [30].

## Acknowledgments

We thank the organizers of the 34th edition of the ICGTMP for their hospitality and the preparation of the congress. We are also glad to have shared the interesting discussions with J.P. Gazeau, M.A. del Olmo, M.A. Lledó, P. Van Isacker, D. Schuch, and F.J. Herranz during the sessions. We thank O. Castaños and E. Perez-Romero for their collaboration in this work.

**Funding information** We thank the support of the Spanish MICINN through the project PGC2018-097831-B-I00 and Junta de Andalucía through the projects UHU-1262561, FQM-381 and FEDER/UJA-1381026. AM thanks the Spanish MIU for the FPU19/06376 predoctoral fellowship.

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
