# Peer review of "Mixed permutation symmetry quantum phase transitions of critical three-level atom models"

_SciPost Physics Proceedings, doi:SciPost Phys. Proc. 14, 036 (2023)_

## Round 1 · Referee Report · Anonymous (Referee 1) · 2023-2-7

Strengths

There is nothing particular strong about this submission. The idea is nice but it's already been published and the current manuscript add nothing to current state of knowledge

Weaknesses

1. The writing is very unclear.
2. The bibliography is incomplete and misleading
3. The motivation for using coherent states in unclear.
4. The treatment of multiplicities is not discussed.

Report

The submission needs considerable revisions before publications.

Requested changes

1. Update the bibliography,
2. Clean up the language
3. Clarify the role of repeated copies of a representation.
4. Clarify how the parity symmetry is broken,
5. Justify the coherent state ansatz, and the parametrization of these states.
6. Clarify the meaning of permutation symmetry in the limit of large N.

Attachment

  • validity: ok
  • significance: low
  • originality: ok
  • clarity: poor
  • formatting: acceptable
  • grammar: below threshold

Author:  Alberto Mayorgas  on 2023-02-17  [id 3371]

(in reply to Report 1 on 2023-02-07)
Category:
correction

Dear Referee,

We carefully address in the attached file the referee's recommendations and comments in
relation to the manuscript scipost\_202212\_00060v1 entitled "Mixed permutation symmetry quantum phase transitions of critical three-level atom models", by A. Mayorgas, J. Guerrero, and M. Calixto, which is submitted to SciPost Physics Proceedings for publication. We have also included the references suggested by the referee. As a result, the length of the article slightly exceeds 8 pages. We hope that this is not a problem.

We thank the referee for their comments and recommendations which helped to improve the article.
I hope that the explanations provided and the changes made to the manuscript make it suitable for publication in SciPost Physics Proceedings.

Looking forward to hearing from you.

Yours sincerely,

The Authors

Attachment:

AnswerToReferee.pdf

---

## Round 2 · Referee Report · Anonymous (Referee 2) · 2023-3-22

Strengths

The content of the submitted work shows a real expertise of the authors in managing permutational symmetries and the study of phase transitions for systems at large number of particles with mixed permutational symmetries. The model considered by the authors is described by the Lipkin-Meshkov-Glick Hamiltonian.
The mathematical aspects are clearly exposed and justified. The numerical explorations are convincing.

Weaknesses

There is no weakness but the fact that the content is quite condensed.

Report

All criteria of the Journal are met in a satisfactory way. The submission excellently proves the strength of symmetry methods in facing with non-trivial questions pertaining with quantum phase transitions.

Requested changes

No change

---

## Editorial Decision

published